# Design considerations for the migration from paper to screen-based media in current health education for older adults: a scoping review

Larissa Taveira Ferraz ,[1] Anna Julia Tavares Santos,[2] Lorena Jorge Lorenzi ,[3] David Mark Frohlich,[1] Elizabeth Barley ,[4] Paula Costa Castro[2]

[1]Department of Music and Media, University of Surrey, Guildford, UK
[2]Department of Gerontology, Federal University of São Carlos, São Carlos, Brazil
[3]Postgraduate Program in Bioengineering, University of São Paulo, São Carlos, Brazil
[4]Mental Health Sciences and Nursing, University of Surrey, Guildford, UK

**Correspondence to**
Larissa Taveira Ferraz;
l.taveiraferraz@surrey.ac.uk

## ABSTRACT

**Objectives** To map the current use of paper-based and/or screen-based media for health education aimed at older people.

**Design** A scoping review was reported following the Preferred Reporting Items of Systematic Reviews and Meta-analyses for Scoping Reviews checklist.

**Data sources** The search was carried out in seven databases (Scopus, Web of Science, Embase, Medline, CINAHL, ACM Guide to Computing Literature, PsycINFO), with studies available from 2012 to the date of the search in 2022, in English, Portuguese, Italian or Spanish. In addition, Google Scholar was searched to check the grey literature. The terms used in the search strategy were older adults, health education, paper and screen-based media, preferences, intervention and other related terms.

**Eligibility criteria** Studies included were those that carried out health education interventions for older individuals using paper and/or screen-based media and that described barriers and/or facilitators to using these media.

**Data extraction and synthesis** The selection of studies was carried out by two reviewers. A data extraction form was developed with the aim of extracting and recording the main information from the studies. Data were analysed descriptively using Bardin's content analysis.

**Results** The review included 21 studies that carried out health education interventions with different purposes, the main ones being promotion of physical activity, hypertension prevention and psychological health. All 21 interventions involved screen-based media on computers, tablets, smartphones and laptops, while only 4 involved paper-based media such as booklets, brochures, diaries, flyers and drawings. This appears to reflect a transition from paper to screen-based media for health education for the older population, in research if not in practice. However, analysis of facilitators and barriers to using both media revealed 10 design factors that could improve or reduce their use, and complementarity in their application to each media type. For example, screen-based media could have multimedia content, additional functionality and interactivity through good interaction design, but have low accessibility and require additional learning due to complex interface design. Conversely, paper-based media had static content and low functionality but high accessibility and availability and a low learning cost.

## STRENGTHS AND LIMITATIONS OF THIS STUDY

⇒ It is the first review that mapped the differences in the use of paper-based and screen-based media for health education for older people.
⇒ The studies were searched in seven databases, in addition to a grey literature search.
⇒ The studies included were those written in Portuguese, English, Spanish and Italian.
⇒ A limitation of the review was having an inclusion criterion that the studies should have a mean age equal to 60 years or older.

**Conclusions** We recommend having improved screen-based media design, continued use of paper-based media and the possible combination of both media through the new augmented paper technology.

**Registration number** Open Science Framework (DOI: 10.17605/OSF.IO/GKEAH).

## INTRODUCTION

The population of older adults aged 60 years and over reached about 14% of the world's population in the year 2020, which means one in seven people. In the year 1950, this percentage was 8% of the population.[1] Population ageing has been triggered by a series of factors, such as a decrease in the fertility rate and an increase in life expectancy.[2] The projection for 2050 is that the number of adults aged over 60 years will reach 22% of the world's population, that is, approximately one in five people.[1] This increase is also leading to more people living with chronic health conditions and a need to improve quality of later life through what is called 'healthy ageing'.

The document entitled 'Decade of Healthy Ageing', by the WHO, proposes a plan to improve the health of the older adult population between 2021 and 2030.[3] It emphasises the importance of health conditions being managed efficiently with health promotion

and disease prevention practices.[3] A key part of this strategy is the provision of effective health communication through media that are disseminated both on paper and screen. According to the WHO, health communication can be defined as 'the use of communication strategies (eg, interpersonal, digital and other media) to inform and influence decisions and actions to improve health'.[4]

Paper-based media are established formats for health education provided by professionals due to their widespread accessibility.[5] Such media were, and still are, widely used in health practices in the form of booklets, leaflets, posters and manuals.[6] In the literature, there are some studies that present recommendations and guidelines for their use, such as the one carried out by Hoffmann and Worrall[55]; they emphasise the importance of these materials as being objective and updated; easy to understand for everyone; using table of contents, subheadings, bullet lists and abstracts; organised, with differentiation between uppercase and lowercase letters and use of bold only for emphasis; using images or drawings only to aid understanding; and using interaction resources, for example, a short questionnaire.

Increasingly, health education materials are being disseminated through applications and websites on screen-based devices such as smartphones and computers.[7] Many of these health education services are targeted at older adults, because they are disproportionately affected by illness and are likely to benefit most from information about healthy ageing practices. It should also be noted that, along with technological advances, there are other situations that enhance the use of digital media, such as the advent of the COVID-19 pandemic with the necessary social distancing.[8] Recommendations on the use and design of screen-based health information are also available in the literature. For example, the study by Zhao *et al*[99] explores the information-seeking behaviour of older adults towards health information and reveals that in addition to personal barriers, there are barriers to using technology, such as unsatisfactory interactivity and usability, inadequate font sizes, dense texts, lack of visual elements and confusing layouts.

Current studies, including reviews, of the use of media in paper or screen formats for health information are carried out separately and seldom compared one with the other. Furthermore, they address a wide range of ages in the participants involved. The aim of this review is to map the use of paper and screen-based media in current health education practices targeted at older people aged 60 years or over. This should allow the identification of the barriers and facilitators to using each medium for this population, comparing the use of these two types of media for health education.

## METHODS AND ANALYSIS

This review was developed according to the guidelines of the Preferred Reporting Items of Systematic Reviews and Meta-analyses for Scoping Reviews[10] and the recommendations of the Joanna Briggs Institute Revisions Manual.[11] The protocol of the review with methodological details is published.[12] The research questions were as follows:

1. What types of paper-based and screen-based media are used in current forms of health education and how are they used for health education of older adults?
2. What are the barriers and facilitators to using paper-based and screen-based media for health information of older adults?
3. What are the differences between the use of paper-based and screen-based media for health education?

### Eligibility, inclusion and exclusion criteria

Scientific articles that had as their objective or central theme a health education intervention carried out with paper and/or screen-based media for health promotion or prevention of older individuals were considered eligible. Studies included were those that carried out or described health education interventions for health promotion or prevention of older individuals using paper and/or screen-based media and that described barriers and/or facilitators to using these media. The studies included should have a sample of subjects with a mean age of 60 years or older and be published in English, Portuguese, Spanish or Italian, from 2012 until the search date. In addition, grey literature studies were included through an additional search in the Google Scholar.

Studies that were not about health promotion or prevention, such as interventions on treating or caring for patients with pre-existing health conditions, were not included. In addition, studies that did not present barriers or facilitators perceived by users in relation to the media were excluded. Political documents, technical reports and studies whose full texts could not be obtained, even after attempts to contact the authors, were excluded.

### Search strategy

A search strategy was developed based on preliminary searches in databases, using the terms *older adults, health education, paper-based media, screen-based media, experience* and *intervention*, in addition to other related terms. The search strategy was adapted using Boolean operators and Medical Subject Headings terms according to each database. The search strategy used in the Scopus database was as follows:

("older person" OR elder* OR "older adults" OR "elderly population" OR "older people" OR ageing OR aging OR "older population" OR geriatric OR "healthy ageing" OR "successful aging") AND ("paper based media" OR "screen based media" OR website* OR platform* OR virtual OR online OR multimodal OR multimedia OR "reading media" OR "digital-based reading" OR "paper based reading" OR flyer OR "media advertisement" OR "print media" OR app OR apps OR tablet* OR smartphone* OR m-health* OR e-health* OR "patient information leaflets") AND ("health education" OR "health information" OR "health communication" OR

"health promotion") AND (preference* OR characteristic* OR experience* OR attribute* OR perception* OR development OR barrier* OR facilitator* OR opportunities OR problem* OR recommendations) AND (trial OR intervention)

Searches were performed in seven databases: Scopus (Elsevier), Web of Science (Clarivate), Medline, Embase (Elsevier), CINAHL (EBSCO), ACM Guide to Computing Literature, PsycINFO (APA) databases. All search strategies used in the databases are described in the online supplemental file. Studies published in Portuguese, English, Spanish and Italian, from 2012 to the date of the search in June 2022, were searched. An additional search on Google Scholar was performed to verify 150 studies according to Google's relevance algorithm in the year 2022. Later, in April 2023, after guidance from the protocol reviewer who suggested the search should include the first 300 studies, based on the paper by Haddaway *et al*,[13] the expansion of the grey literature was carried out, updating the search for April 2023, until the first 300 studies from Google Scholar were identified.

### Study/source of evidence selection

The selection of studies was carried out in five stages, starting with the search in the databases in which the identified studies were extracted into the Excel program and duplicates were removed. The next step was a pilot test in which two reviewers checked 25 random studies found in the search according to the eligibility criteria, obtaining a 92% agreement rate. The third step was the reading of abstracts, titles and keywords of all studies by two reviewers independently following the eligibility criteria. The fourth step was the reading of the full texts of the studies accepted in the previous step by the reviewers independently according to the inclusion criteria. In both phases, a third reviewer resolved disagreements. The last step was the additional strategy, which was the reading of the first 300 studies found in Google Scholar according to the relevance algorithm.

### Data extraction

Data from the studies included in the review were extracted into a table developed and checked by the authors in three studies. The extracted data contained information about the studies such as authors' names, title, year of publication, country of origin and methods used, information about the sample such as size, gender and mean age, characteristics of the intervention such as duration and frequency, health condition verified, paper-based media used, screen-based media used, barriers to using paper and/or screen media and facilitators to using paper and/or screen media. The extraction table was described in the protocol for this scoping review; however, the topic 'Is the health education aimed at the caregiver?' was withdrawn. This topic was removed from the extraction table, considering that throughout the study, it was identified that the papers included were mostly focused on the participants 60+ years and not their caregivers.[13]

### Data analysis and presentation

The data extracted from the included studies were presented in a narrative form and through tables and the Preferred Reporting Items for Systematic Reviews and Meta-Analyses flow chart. Quantitative data were analysed using relative and absolute frequency, and mean and SD. Qualitative data about the barriers and facilitators to using paper and/or screen-based media were analysed using Bardin's content analysis.[14] Barriers and facilitators were grouped into categories according to their similarities.

## RESULTS

The databases searched resulted in 1392 studies, as shown in the flow diagram in figure 1. Duplicate studies were removed which resulted in 1080 articles found. All the studies found were screened by two reviewers, and based on the eligibility criteria, a total of 134 articles were sent to the reading phase of the full texts. In the selection of studies based on the inclusion criteria, the total number of included studies was 15, and 119 were excluded as described in the flow chart (figure 1). In the additional strategy, which was the reading of the 300 studies found in Google Scholar in order of relevance, it was identified that 21 were duplicates that had already been found in the search of the databases. Thus, in order to have 300 studies without duplicates, 321 studies were screened. As a result, 95 studies were selected for full reading and, subsequently, 6 were included based on the same inclusion criteria applied to full texts. Therefore, the main search included 15 studies and the Google Scholar search 6 studies; this review then included a total of 21 studies.

### Characteristics of the studies

Descriptive characteristics of the included studies (n=21), such as year of publication, country where it was carried out, year of data collection, methodology, intervention, sample size and average age of participants, are described in table 1.

Regarding the year of publication described in table 1, approximately 61.9% of the articles were published in 2017, 2018, 2019, 2021 or 2022. Respectively, the USA, Australia, The Netherlands, New Zealand and South Korea were the countries where most studies originated in this review. The year of data collection in the studies showed great variability, with the oldest year being 2007 and the most recent year being 2021. The majority of studies were carried out longitudinally (80.95%).

The interventions were mostly face-to-face (52.38%) and had a minimum duration of 30 min and a maximum duration of 6 months. The studies' sample ranged from 5 to 703 participants; however, most had 31–60 participants (33.33%) or 5–30 participants (28.57%), and most of the studies had participants with an average age between 65 and 69 years old (33.33%) or 60 and 64 years old (23.81%).

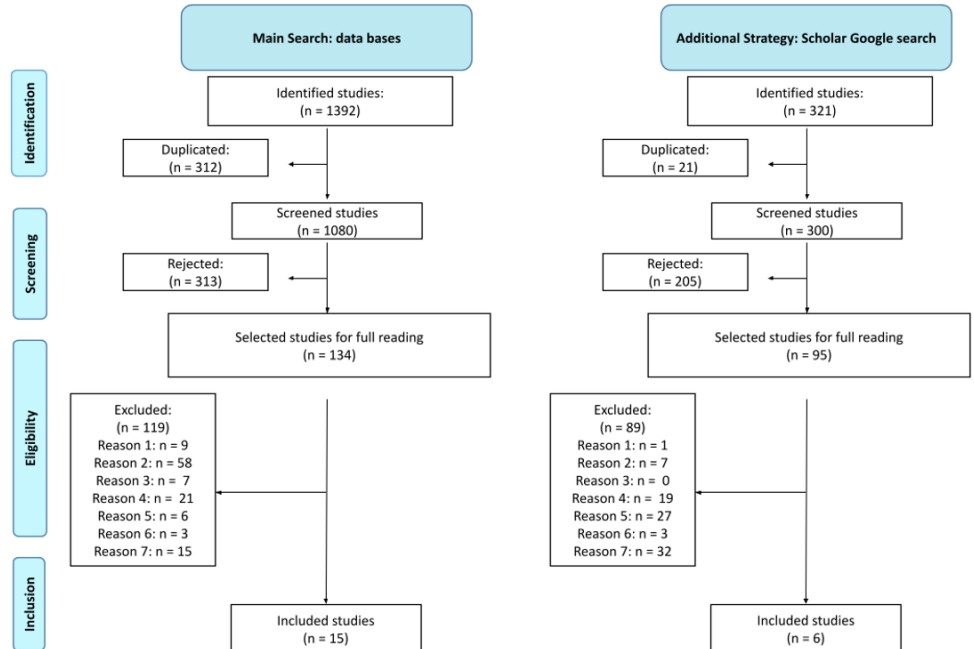

**Figure 1** Preferred Reporting Items for Systematic Reviews and Meta-Analyses flow diagram of the studies included in the review that presents screen-based and/or paper-based media for health education. Reason 1: it is a protocol, literature review or has only an abstract available; reason 2: it does not present the mean age or the mean age is less than 60 years; reason 3: it does not have participants; reason 4: it does not have a health education intervention or the health education is not for promotion or prevention; reason 5: health education does not use paper or screen-based media; reason 6: it does not describe the health education; reason 7: it does not present barriers or facilitators perceived by users in relation to the media used. Source: developed by the authors.

Two studies (9.52%) did not report the mean age but did not have adults (aged 18–59 years) in the sample. The mean total age of participants across all included studies was 68.51 (±5.06) years. Most studies (90.48%) were performed with individuals of both genders, and the other studies did not report data on gender. In addition, 47.62% of the studies did not have adults (aged 18–59 years) in the sample, 47.62% of the studies had adults and one study did not report this information, but the mean age was over 60 years.

### Health education themes and types of media found in the studies

In response to the first research question of this review about the format of health information used and for what purposes, table 2 presents all the studies included in this scoping review, with their authors, year of publication, central theme of the health education and what type of digital/paper-based media was used.

The central theme of health education had great variability among the included studies; however, it was identified that the themes that most appeared were physical activity promotion (14.29%), psychological health (14.29%) and hypertension prevention and care (9.52%).

Among the screen-based media formats used for health education in the studies, the following media were identified: websites (61.90%), applications (apps) (23.81%), emails (9.52%) and animations on Microsoft PowerPoint (4.76%). These media were used on the devices such as computers (52.38%), tablets (28.57%), smartphones (19.05%) and laptops (4.76%). Some studies used more than one medium or device. Regarding the paper-based media formats used for health education in the studies, it was identified that they were booklets (A4 paper), brochures, handouts, diaries, flyers, drawings and printed information (without specification).

A major finding of the review was the imbalance of studies on paper and screen-based interventions. All 21 articles included in this study featured screen-based media, and among these, only 4 reported the contrasting use of paper-based media. No studies were found that reported on the use of paper-based interventions alone.

### Categorisation of barriers and facilitators

In response to the second research question about the barriers and facilitators to media usage, it was identified that the vast majority were about screen-based media. Barriers considered were difficulties that older people had when using screen or paper-based media, factors that hindered the use of the media or the lack of features of the media. Facilitators, on the other hand, were factors that contributed or motivated individuals to use the media. Thus, they were grouped according to their similarity into 10 categories and described below:

► Accessibility—the diversity of users able to access the media system.
► Availability—the availability of the media system.
► Content design—the design of the media content itself.
► Functionality—what the media system does.

| Table 1 Descriptive characteristics of the included studies | | |
|---|---|---|
| Variables | n | % |
| Year of publication | | |
| 2012 | 1 | 4.76 |
| 2013 | 1 | 4.76 |
| 2014 | 2 | 9.52 |
| 2015 | 2 | 9.52 |
| 2016 | 2 | 9.52 |
| 2017 | 2 | 9.52 |
| 2018 | 5 | 23.81 |
| 2019 | 2 | 9.52 |
| 2021 | 3 | 14.29 |
| 2022 | 1 | 4.76 |
| Location: country | | |
| USA | 5 | 23.81 |
| Australia | 3 | 14.29 |
| The Netherlands | 3 | 14.29 |
| New Zealand | 2 | 9.52 |
| South Korea | 2 | 9.52 |
| Belgium | 1 | 4.76 |
| Malaysia | 1 | 4.76 |
| Germany | 1 | 4.76 |
| Taiwan | 1 | 4.76 |
| International (several countries) | 2 | 9.52 |
| Year of data collection | | |
| 2007–2009 | 1 | 4.76 |
| 2010–2011 | 1 | 4.76 |
| 2012 | 2 | 9.52 |
| 2014–2015 | 1 | 4.76 |
| 2015 | 1 | 4.76 |
| 2016 | 1 | 4.76 |
| 2017 | 1 | 4.76 |
| 2017–2018 | 1 | 4.76 |
| 2020 | 1 | 4.76 |
| 2020–2021 | 1 | 4.76 |
| NR | 10 | 47.62 |
| Methodology | | |
| Longitudinal | 17 | 80.95 |
| Cross-sectional | 4 | 19.05 |
| Intervention | | |
| In person | 11 | 52.38 |
| Online | 2 | 9.52 |
| In person and online | 8 | 38.10 |
| Sample size | | |
| 5–30 | 6 | 28.57 |
| 31–60 | 7 | 33.33 |
| 84 | 1 | 4.76 |
| Continued | | |

| Table 1 Continued | | |
|---|---|---|
| Variables | n | % |
| 121–150 | 4 | 19.05 |
| 204 | 1 | 4.76 |
| 218 | 1 | 4.76 |
| 703 | 1 | 4.76 |
| Average age of participants | | |
| 60–64 years | 5 | 23.81 |
| 65–69 years | 7 | 33.33 |
| 70–74 years | 4 | 19.05 |
| 75–79 years | 3 | 14.29 |
| Unknown | 2 | 9.52 |

Source: developed by the authors.
NR, not reported.

- ► Interaction design—design of the sequential organisation of the interaction with the media system.
- ► Interface design—design of the input and output interfaces.
- ► Learning—the ease of learning to use the media system.
- ► Privacy—privacy of personal data in the media system.
- ► Social interaction—the support of interaction with other people around the media system.
- ► Usability—the ease of using the media system.

The topics defined for the categorisation of barriers and facilitators seem to be the design aspect of media systems in general that can have both negative (barriers) and positive (facilitators) effects. There are two sides to the same coin because if the media are done well, they can improve health education, but if done badly, they can lower it. The categories found in barriers and facilitators related to screen-based and paper-based media are described in table 3. The table describes the number of studies that identified each category, with some studies finding more than one type of category as barriers and/or facilitators. In the following subsections, we go through the cited barriers and facilitators for each medium.

### Barriers to using screen-based media
The main categories that appeared as barriers to using screen-based media were learning (eg, lack of computer training, lack of instructional videos, lack of clear guidelines, in-person technology training required for older adults), accessibility (eg, preference for a larger screen using a tablet or laptop, preference for hard copies of the information, because 'a sad thing is you can't touch her or talk to a real person'), content design (eg, strange content, technical terms, no up-to-date information, no varied levels of information), functionality (eg, lack of reward, need to address some technical improvements like the function to delete messages, difficult to define a goal, lack of an icon to facilitate printing) and usability (eg, indirect assignments that required big effort, difficult

**Table 2** Studies included in the review, health education provided and types of paper-based and/or screen-based media used

| Article title | Authors, year | Central theme of the health education | Screen-based media format | Paper-based media format |
|---|---|---|---|---|
| A Digital Human for Delivering a Remote Loneliness and Stress Intervention to At-Risk Younger and Older Adults During the COVID-19 Pandemic: Randomized Pilot Trial[35] | Loveys et al, 2021 | Mental health and well-being | Computer, tablet or smartphone (website) | NA |
| A Multi-faceted Approach to Promote Comprehension of Online Health Information Among Older Adults[36] | Chin et al, 2018 | Self-care regarding hypertension | Computer | NA |
| Acceptability of a guided self-help Internet intervention for family caregivers: Mastery over dementia[37] | Pot et al, 2015 | Psychological distress of family caregivers of people with dementia | Computer (website) | NA |
| Assessing the Impact of a Game-Centered Mobile App on Community-Dwelling Older Adults' Health Activation[38] | Crandall et al, 2019 | Risk of falls and osteoarthritis | Tablet (app) | NA |
| Co-design of an evidence-based health education diabetes foot app to prevent serious foot complications: a feasibility study[39] | Ogrin et al, 2018 | Foot health for people with diabetes | Smartphone (app) | NA |
| Development and assessment of a web-based intervention for educating older people on strategies promoting healthy cognition[22] | Vanoh et al, 2018 | Precautionary strategies for mild cognitive impairment | Computer (website) | NA |
| Development and Evaluation of a Computerized Multimedia Approach to Educate Older Adults about Safe Medication[40] | Im and Park, 2014 | Safe medication | Computer (website) | NA |
| Development and Validation of an Interactive Internet Platform for Older People: The Healthy Ageing Through Internet Counselling in the Elderly Study[23] | Jongstra et al, 2017 | Improving the cardiovascular risk profile of old people with high cardiovascular risk | Computer (website) | NA |
| Development of a communication intervention for older adults with limited health literacy: Photo stories to support doctor–patient communication[17] | Koops et al, 2016 | Doctor–patient communication and communicative health literacy | Computer (website) | Booklet (A4 paper) |
| Differences in the use and appreciation of a web-based or printed computer-tailored physical activity intervention for people aged over 50 years[18] | Peels et al, 2013 | Physical activity promotion | Computer (website; email) | Printed information, brochures and handouts |
| Efficacy of a web-based, center-based or combined physical activity intervention among older adults[19] | Mouton and Cloes, 2015 | Physical activity | Computer (website; email) | NA |
| Evaluation of the usability and acceptability of the InnoWell platform as rated by older adults: Survey study[41] | LaMonica et al, 2021 | Biopsychosocial domains | Tablet, computer, laptop, smartphone (website) | NA |
| Improving older adults' e-health literacy through computer training using NIH online resources[42] | Xie, 2012 | Improving the ability of seniors to search, find and understand health information online | Computer (website) | NA |
| Mobile App Use for Insomnia Self-Management in Urban Community-Dwelling Older Korean Adults: Retrospective Intervention Study[25] | Chung et al, 2018 | Sleep education | Smartphone (app) | NA |

Continued

**Table 2** Continued

| Article title | Authors, year | Central theme of the health education | Screen-based media format | Paper-based media format |
|---|---|---|---|---|
| Self-efficacy, self-regulation, and physical activity behavior in type 2 diabetes[29] | Olson, 2014 | Metabolic health education | Computer (website) | NA |
| The Application of Virtual Reality in Patient Education[16] | Pandrangi et al, 2019 | Health education—health status | Computer (website) | Flyers, drawings and brochures |
| The GezelschApp A Dutch Mobile Application to Reduce Social Isolation and Loneliness[43] | Jansen-Kosterink et al, 2018 | Social isolation and loneliness among old people | Computer and tablet (app) | NA |
| The Impact of 3-D Models vs Animations on Perceptions of Osteoporosis and Treatment Motivation: A Randomised Trial[44] | Jones et al, 2017 | Information about osteoporosis | Tablet (animations on Microsoft PowerPoint) | NA |
| The influence of robot-assisted learning system on health literacy and learning perception[45] | Wei et al, 2021 | Hypertension prevention for older adults | Tablet and computer | NA |
| Web-based oral health promotion program for older adults: Development and preliminary evaluation[46] | Mariño et al, 2016 | Oral health promotion | Computer (website) | NA |
| Web-based vs print-based physical activity intervention for community-dwelling older adults: crossover randomized trial[15] | Pischke et al, 2022 | Promoting a physically active lifestyle | Computer (website and app) | Brochures and diary |

Source: developed by the authors.
app, application; NA, not applicable.

passwords provided to log in for the first time). Moreover, the following barriers were also identified: availability (eg, computer availability and computer reserves), interaction design (eg, aspects participants felt could be improved were the human-like interaction like behaviours and voice and the conversation design), interface design (eg, videos were not easy to see because they had small size, small font size, and lack attractive photos and project colours), privacy (eg, participants did not post on the discussion board due to privacy concerns) and social interaction (eg, limited app interaction with participants).

### Facilitators to using screen-based media
The main categories that appeared as facilitators to using screen-based media were content design (eg, information with revised content from researchers and experts that helped to understand the content; easy-to-understand text structures, bullet lists and phrases; animation based

**Table 3** Barriers and facilitators found in the included studies

| Topics | Screen-based media | | Paper-based media | |
|---|---|---|---|---|
| | Barriers | Facilitators | Barriers | Facilitators |
| Accessibility | 5 | 3 | — | — |
| Availability | 1 | 3 | — | 2 |
| Content design | 4 | 8 | — | 1 |
| Functionality | 3 | 6 | 2 | — |
| Interaction design | 2 | 4 | — | — |
| Interface design | 2 | 4 | — | — |
| Learning | 6 | 4 | — | — |
| Privacy | 1 | — | — | — |
| Social interaction | 1 | 2 | — | — |
| Usability | 3 | 6 | — | — |

Source: developed by the authors.

on the life story of an older adult as the main character contributed to enhancing the self-efficacy of the media for the participants), usability (eg, operating the program just by clicking the mouse was easy, it was simple and easy to navigate on a mobile phone, easy to use the interface, most participants found the sites easy or very easy to use), functionality (eg, participants really liked the function of being able to make new friends and send and receive messages, and also mentioned the videos as the most useful component, in addition to feedback being helpful), accessibility (eg, the app was appropriate for people with a low level of education, the touch screen helped older adults navigate the app) and interaction design (eg, participants appreciated the interactive feature of the platform like the interactive videos and interactive questions that helped with problem-solving and how it improved their interpersonal skills). In addition, the following facilitators were also identified: availability (eg, information on the smartphone can be seen at any time quickly, websites can be permanently accessed), interface design (eg, the animations on the tablet allowed more of the information to be visualised, participants liked the appearance, cross-referencing a number of the images and statements from various sections was easy), learning (eg, participants reported that they easily learnt how to use the application, they also enjoyed and felt comfortable with self-learning), social interaction (eg, many participants expressed how much they enjoyed the social aspect of the app, and considered the app a pleasant and safe way to find new friends).

### Barriers to using paper-based media

Regarding the barriers related to paper-based media, the study by Pischke et al[15] found functionality to be a barrier as the printed version seemed to be impractical with regard to physical activity. Pandrangi et al[16] also identified functionality as a barrier since participants perceived the paper format as a traditional method and considered virtual reality more useful.

### Facilitators to using paper-based media

In the study by Koops et al,[17] it was identified that availability and content design were the facilitators to using paper-based media, as the participants reported that they could take them anywhere and they liked that the stories were similar to real life and that it was easier for them to understand. Peels et al[18] also found availability to be a facilitator since the participants had more intentions to keep the information and to use the concepts learnt in the future.

### Differences between the use of paper-based and screen-based media for health education

Regarding the third research question about the differences between the use of paper-based and screen-based media, this was difficult to answer because of the small number of direct comparisons made. However, three particular barrier/facilitator categories appeared to relate to the contrasting properties of paper and screen-based media, as expressed in the four studies using both paper and screen-based media. These were availability, functionality and learning.

Availability appeared in two studies as facilitators concerning the use of paper-based media because of actually having the paper and holding it with hands and being able to use it at any time. For the use of screen-based media, there is a need for more effort, for example, when having to turn on a computer, log in to a website, download an application, etc, in order to be able to start using these media.[17 18]

Considering the use of paper-based media, two other studies identified functionality as a barrier to their use, due to the fact that their use is limited compared with screen-based media, in which there is the possibility of having different functions, such as having feedback and interactive videos, setting goals and being able to make new friends.[15 16]

In addition, the learning category is related to the difference in the use of paper and screen-based media due to the older participants who have difficulty using screen-based media, as they have no affinity with them and need training to use them.[15 19] Thus, for individuals with difficulty in relation to the use of screen-based media, the use of paper-based media is preferable, as seen in the study by Pischke et al.[15]

### DISCUSSION

The review identified 21 studies that carried out health education interventions to promote ways of staying healthy longer to an ageing population subject to more chronic health conditions; the main subjects in the interventions were related to promotion of physical activity, hypertension prevention and psychological health. This was reflected in the health education subjects of the papers, which were directed to issues such as metabolic health, sleep, healthy eating, physical activity and cognition, and focused on the development and maintenance of autonomy and independence, enabling healthy ageing, according to the WHO definition.[3]

The interventions were done through the use of screen-based media such as websites, apps and emails, on desktop computers, tablets, smartphones and laptops, and paper-based media such as booklets, brochures, diaries, flyers and drawings. We found that the barriers and facilitators to using these media were essentially design aspects that could improve media usage if done well or reduce usage if done badly. This led us to propose a taxonomy of 10 design features and considerations which apply to both media, somewhat unequally. The main design considerations for the use of screen-based media were related to content design, functionality, usability, accessibility and learning. On the other hand, the main design considerations for the use of paper-based media were availability and content design, but such media were often considered traditional because they did not have extra functionality.

We can see from the diversity of paper and screen-based media in the studies that these terms each refer to quite different formats. In fact, they gloss these differences and do not really do justice to the different interactions people could have with each format. This may be one reason for the ambiguity of findings in such comparisons in the past, and suggests greater attention to *particular* paper and screen formats in future research. For example, screen-based information was presented on four physical devices in the above studies, including desktop and laptop computers, tablets and smartphones. Sometimes, this information was on a website or presented in a more interactive app, leading to a matrix of eight possibilities across devices with different user experiences. On the other hand, paper-based information was presented on single-page leaflets, flyers and handouts, as well as multipage booklets or diaries. Each of these has different affordances for reading, finding, handling and capturing information, and is not equivalent to favourable user experiences. Further work is required to understand the subtleties of design, interaction and experience of using all these 'paper' and 'screen' formats, since the current papers do not go into sufficient detail.

Regarding the types of digital media used for health education, the WHO has a 'Global Strategy on Digital Health 2020–2025'[20] with the purpose of strengthening health systems through digital health technology, and this will engage patients to achieve the vision of health and well-being for all. This strategy signals that practices for public awareness of digital health must be addressed, which includes health literacy and patient engagement.[20] Our findings clearly reveal a noticeable trend towards digitisation and a preference for screen-based media, as shown in the literature. When analysing the review data from the last decade, we observed a significant increase in the number of studies and publications focusing on screen-based media across all years, as depicted both in tables 1 and 2. Our study found barriers and facilitators that, if properly addressed, will make new digital health strategies more user-friendly for older people, which will improve the engagement of this group in health education interventions and consequently improve health literacy. Key among these were the *functionality* and *content*, which were shown to have a great impact, both positive and negative, on interventions.

Based on the findings of our review, the discussion in the literature emphasises the significance of interface design and screen media usability. Morey et al[21] identified issues with regard to excessive icons and low-contrast screen, highlighting participant concerns. Vanoh et al's[22] study underscored that older users prefer visually appealing media featuring attractive photos and colours, while Jongstra et al[23] revealed older users' need for different functionalities. Hence, it is imperative incorporating diverse functionalities tailored to the requirements of the older population, coupled with an engaging design characterised by colour contrast and intuitiveness to enhance usability.

The adoption of a user-centred design approach is crucial to meet the unique needs of older people. Co-design sessions, as suggested by Leme,[24] serve as an effective strategy for involving older individuals in the creation and design processes of new tools across various media. In this way, participatory design enables the creation of intuitive media, which, consequently, older individuals will find easy to use. Moreover, it was identified in the included studies that one of the main barriers was the difficulty of learning these screen-based media. Therefore, it is crucial that public policies are developed and improved for the digital inclusion of older people, such as programmes that teach these individuals how to use these media, so that they can enjoy all the benefits of digital technologies. The study by Chung et al,[25] which aims to analyse the usability of a sleep education app for older Koreans, presents the users' ease of learning to use the application as a perceived facilitator.

Regarding screen-based media use, learning to use these media was one of the most prevalent barriers. According to De Paiva and Alves,[26] the older population today are needed to be accompanied when they need to use the internet, printers and computers. Nowadays, with the technological evolution, with games with different levels of proficiency and the emergence of artificial intelligence, the older population finds it difficult to adapt.

In one of the studies of this review, by Pischke et al,[15] that compared the acceptance and effectiveness of two interventions for physical activity promotion for older adults through websites, apps, brochures and diaries, they found individuals who have no affinity with screen-based media preferring paper-based media. In another study found in this review, by Peels et al,[18] based on usability as a barrier, they reported that 'the assignments that had not been integrated into the web-based tailored advice required greater effort to participate in, therefore it is notable that not being intuitive leads to a greater amount of effort. Thus, older people have greater difficulty with screen-based media and require more training. This training requirement is urgent at the current moment, given that the younger population, who are the future older people, are already growing up with skills and will likely have an easier time using these media.[18]

Additionally, functionality is identified as a barrier to using e-health for older adults, as noted in Wilson et al's[27] scoping review. On the other hand, ease of use emerged as a facilitator, as observed in both Alruwaili et al's[28] and Wilson et al's[27] studies, aligning with our usability review category.

Our study's results align with concerns mentioned in Olson's[29] research, particularly regarding privacy, which is also identified as a barrier in related studies such as Alruwaili et al's[28] systematic review and Wilson et al's[27] study. Privacy and data security are recognised as obstacles to digital health interventions among older adults. Therefore, it is essential to create public policies that ensure the privacy and security of users' data when using screen-based media in health education interventions.

It is also important that policies regarding screen-based media are presented to the user at the beginning of the interaction with the media, as this will inform and ensure safety for them.

Digital engagement carries the danger and opportunity of data capture and user profiling. The Global Strategy on Digital Health 2020-2025[20] emphasises that health data are sensitive and personal; thus, they require a high standard of security and protection. Lack of trust and confidence in divulging personal health data can discourage people from fully participating in health education interventions. The WHO mentions the need for a strong legal and regulatory foundation to protect privacy, integrity and confidentiality and deal with data cyber security. It is also important to maintain transparency and communicate effectively about data security strategies to users.[20]

Regarding the use of paper-based media for health education, according to the findings of this review, it is clear that these media are still very much alive today, as revealed in the study by Koops *et al*[17] who used a website and a booklet to develop a doctor–patient communication for older adults with limited health literacy. In this study, a booklet with photographic stories that aided in the communication between the doctor and patient was considered to have good availability, as it could be given out and used anywhere. This is likely to reflect a pervasive practice of displaying posters and giving out leaflets in general practitioner surgeries and hospitals across the globe, regardless of their lack of representation in this research review. Indeed, this is mentioned in the wider literature, such as a study by Maskell *et al*,[30] which found that more than three-quarters of patients noticed printed materials in the waiting rooms of general clinics in England, and a large proportion of them read these materials. Hence, paper-based media are already widely accepted and available as part of modern health education practices.

Regarding the international reach of these studies, we note that they come largely from high-income countries as shown in table 1. There is a lack of research and representation from low-income and middle-income countries, especially those with vulnerable populations, and large countries with vast and remote regions facing challenges in accessing healthcare services, as shown in table 1. For populations with limited education and financial vulnerability, paper-based materials appear more accessible due to their traditional usability and cost-effectiveness. However, for individuals who are actually illiterate, screen readers and audio content, for instance, offer a level of accessibility that paper-based materials cannot match. These functionalities are facilitators found in screen-based media seen in this review, as videos were mentioned as the most useful component in the study of Olson.[29] Moreover, advancements in satellite technologies are now integrating previously isolated areas into the global network, potentially granting access to health information for a substantial number of people who currently lack access to up-to-date pamphlets and leaflets. This integration and

interactivity brought about by screen-based media also enable these individuals to connect with health teams and professionals in more densely populated areas. This, in turn, could lead to more personalised and humanised health education guidelines and interventions through technology. Overall, these developments hold promise in bridging the healthcare information gap and fostering improved health outcomes for underserved communities.

Given the advantages and disadvantages of paper and screen-based information emerging from this review, it may be valuable to combine them in future health campaigns, perhaps at a new level of granularity. This can be done through *augmented paper* technology which supports the reading of printed hotlinks on a nearby smartphone.[31] This results in a paper-and-screen reading experience in which users move back and forth between media which have been carefully designed to complement each other. For example, a printed travel guide can play interviews with local characters, video walks through the landscape, and trigger phone calls to organisations or show updated timetable information.[32] The user in this scenario does not have to know how to use the web to access the digital information. They simply take a picture of the printed page, speak its page number, turn the page or move a bookmark to access the links.[33] A similar approach could be taken to the design of augmented patient information leaflets, posters and cards, which could play multimedia illustrations, spoken translations or instructions, and call local clinics or medical services for more advice.[34]

This review is limited to studies reporting the use of participants with a mean age equal to 60 years or older. Thus, studies that had not reported the mean age or age range of participants were excluded, but they might have been included if they had reported this information. Another limitation we encountered was the scarcity of studies exclusively using paper-based media, as none were solely dedicated for this form, and only 4 out of 21 studies contrasted paper with screen-based media. Following the completed review, we identified 'printed material' as a potential descriptor that other researchers can use. While this trend might indicate the increasing use of screen-based media, it limits the enhancement of comparisons between the barriers and facilitators of both. In addition, a methodological bias was the inclusion criterion that participants should have reported barriers or facilitators, which resulted in a reduction in the number of articles included. Furthermore, as it is a scoping review, its objective is to map all studies on the topic; therefore, an assessment of the quality of the included studies was not carried out.

The strengths of this review are as follows: this is the first study that identifies and compares digital and paper-based media for health education aimed at older people, as well as the design considerations related to their use; studies were searched in seven databases and in the grey literature; studies searched were in four languages, namely Portuguese, English, Spanish and Italian; and

systematic methods were used for the selection of studies, for example, all titles, abstracts and full texts were screened by two reviewers independently.

## CONCLUSION

A transition or migration appears to be taking place in the provision of health information from paper to screen. In this review, we have examined recent literature on the design and use of such information for older people across these formats and found an over-representation of research on screen-based interventions. Although this suggests that the future lies in greater digital engagement, a careful reading of 21 papers shows many barriers to the use of screen-based information and corresponding facilitators to the use of paper-based information for health, especially for an older population and a broader international context. We therefore recommend greater attention to the barriers to using screen-based information across different devices and services, continued use of paper-based media and the possible combination of both media through new augmented paper technology.

**Contributors** All authors have made substantial contributions to the development of the manuscript. LTF and AJTS conceptualised the research question and prepared the first draft of the manuscript. LTF, AJTS, LJL, PCC, DMF and EB contributed to the refining of the study design and guided the review development. LTF, AJTS and LJL analysed the results. PCC, DMF and EB contributed to the discussion. LTF, AJTS, LJL, PCC, DMF and EB contributed to the writing and revising this review. DMF assisted in the English language editing. LTF is the author acting as guarantor.

**Funding** LTF received funding from the University of Surrey Doctoral College (Breaking Barriers Studentship Award). AJTS received funding from São Paulo Research Foundation (FAPESP, no 2022/08918-6) for the development of this review. LJL received funding from Coordination for the Improvement of Higher Education Personnel (CAPES, Financial Code 001).

**Competing interests** None declared.

**Patient and public involvement** Patients and/or the public were not involved in the design, or conduct, or reporting, or dissemination plans of this research.

**Patient consent for publication** Not applicable.

**Ethics approval** Ethical approval is not applicable to this scoping review, as it only gathered information from published literature.

**Provenance and peer review** Not commissioned; externally peer reviewed.

**Data availability statement** No data are available.

**ORCID iDs**
Larissa Taveira Ferraz http://orcid.org/0000-0003-3582-8097
Lorena Jorge Lorenzi http://orcid.org/0000-0002-2378-1263
Elizabeth Barley http://orcid.org/0000-0001-9955-0384

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
