## [Reviewer comments · BMJ Open]

ARTICLE DETAILS

TITLE (PROVISIONAL)	Design considerations for the migration from paper to screen media in current health education for older adults: A scoping review
AUTHORS	Ferraz, Larissa; Santos, Anna Julia; Lorenzi, Lorena; Frohlich, David; Barley, Elizabeth; Castro, Paula

VERSION 1 – REVIEW

REVIEWER	Shahar, Suzana Universiti Kebangsaan Malaysia, Faculty of Health Sciences
REVIEW RETURNED	28-Sep-2023

GENERAL COMMENTS	This review is a few of its kind comparing both paper and screen based health education among older adults. A few recommendations for the authors to improve the manuscript is as follows:- 1. Page 8, last para on the exclusion criteria to be shifted to methods section2. Page 8, second last para, it is suggested to describe clearly for sampling of the second search, ie. google search, that end up with 6 articles, and further adding to the main search of 15 articles3. Limitation, ie, the limited of paper based research need to be expanded further, and how this influence the accuracy of the findings and conclusion4. Why accessibility of screen based has been reported as low compared to paper based need to be further expanded, with inclusion of factors such as digital illiteracy, particularly in the countries or regions where there is issues on digital accessibility etc.
--

REVIEWER	Okpalauwaekwe, U University of Saskatchewan, Medicine
REVIEW RETURNED	18-Oct-2023

GENERAL COMMENTS	It was a pleasure reviewing and learning about your work. The article is well-structured and provides a comprehensive review of the use of paper and screen-based media for health education among adults aged 60 and above. The authors clearly outlined the methodology, including the search strategy, data extraction, and analysis. The results are presented in a clear and concise manner, and the discussion provides a thoughtful interpretation of the findings. However, there are several areas where the article could be improved. Find below my general and specific comments: General Comments
--

	1. The article could benefit from a more explicit statement of the research question or objective at the beginning of the paper. While it is implied that the paper is examining the use of paper and screen-based media for health education among older adults, a clear statement of the research question would help to guide the reader. 2. Authors have done a good job of describing their search strategy and data extraction process. However, they could provide more detail on how they assessed the quality of the included studies. For example, did they use a standardized tool for quality assessment? This would add to the rigour of the research. 3. The discussion section could benefit from a more in-depth interpretation of the results. The authors could consider discussing the implications of their findings for practice and policy, and how their results compare to previous research in this area. Specific Comments Line 7-8: "The extraction table was described in the review protocol, however there was a change regarding the topic "is the health education aimed at the ... caregiver?" which was withdrawn, as the target population of the review study is adults 60+ regardless of occupation." This sentence is a bit confusing and could be clarified. It would be helpful to explain why this change was made and how it impacted the review. Line 18-19: "In the additional strategy, which was the reading of the 300 studies found in Google Scholar in order of relevance, it was identified that 21 were duplicates that had already been found in the search of the databases, thus 321 studies were identified in order to have 300 studies without duplicates to be screened for eligibility." This sentence is quite long and could be broken up into two sentences for clarity. Line 34-35: "The younger generations, such as those born in 1980-2010, grew up adapting to new technologies, as they already existed when they were born." This sentence seems out of place in the paragraph and doesn't add to the discussion of the results. Consider removing or rephrasing. Line 46-47: "Indeed, this is mentioned in the wider literature. For example Maskell, McDonald and Paudyal (2018)³⁸ studied the use of printed materials used in the waiting rooms of general clinics in England and found that they were noticed by more of three-quarters of patients, and read by a large proportion of them." This sentence could be rephrased for clarity. Consider something like: "This is supported by wider literature, such as a study by Maskell, McDonald and Paudyal (2018), which found that more than three-quarters of patients noticed printed materials in the waiting rooms of general clinics in England, and a large proportion read them" The authors could provide more detail on how they categorized the barriers and facilitators to use. What criteria did they use to determine what constitutes a barrier or facilitator? This would add to the rigour of the research The authors could provide more context when discussing the studies included in the review. For example, when discussing the
--	---

	study by Chung et al., 2018, it would be helpful to provide a brief summary of the study's findings and how they relate to the overall research question. The authors could provide more analysis on the implications of the barriers and facilitators identified. How do these barriers and facilitators impact the use of paper and screen-based media for health education among older adults? This would add depth to the discussion section Line 72-73: "Mobile App Use for Insomnia Self- Management in Urban Community- Dwelling Older Korean Adults: Retrospective Intervention Study 28 Chung et al., 2018 Sleep education Smartphone (app) NA." This sentence could be clarified. It would be helpful to provide a brief summary of the study's findings and how they relate to the overall research question. Line 78-79: "The Application of Virtual Reality in Patient Education³⁰ Pandrangi et al., 2019 Health education - health status (Website) Flyers, drawings and brochures." This sentence could be broken up for clarity. Consider something like: "Pandrangi et al., 2019 conducted a study on the application of virtual reality in patient education. The study used a website, flyers, drawings, and brochures for health education" Line 84-85: "The Impact of 3-D Models versus Animations on Perceptions of Osteoporosis and Treatment Motivation: A Randomised Trial³² Jones et al., 2017 Information about osteoporosis Tablet (animations on Microsoft PowerPoint) NA." This sentence could be rephrased for clarity. Consider something like: "Jones et al., 2017 conducted a randomized trial studying the impact of 3-D models versus animations on perceptions of osteoporosis and treatment motivation. The study used a tablet with animations on Microsoft PowerPoint" Line 90-91: "Categorization of barriers and facilitators In response to the second research question about the barriers and facilitators to use, it." This sentence is incomplete and could be clarified. It would be helpful to provide more detail on how the authors categorized the barriers and facilitators to use. Overall, it is a valuable contribution to the literature on a pertinent topic with very good quality points. Only minor revisions to expand some result details.
--	--

VERSION 1 – AUTHOR RESPONSE

Considering Reviewer 1, Prof. Suzana Shahar, Universiti Kebangsaan Malaysia:

We appreciate the following comment from the reviewer “This review is a few of its kind comparing both paper and screen based health education among older adults”.

About the comment “Page 8, last para on the exclusion criteria to be shifted to methods section”

Answer: The last paragraph on page 8 was removed and these exclusion criteria were described in the method section in the following paragraph "Studies that were not about health promotion or

prevention, such as interventions on treating or caring for pre-existing health conditions were not included. In addition, studies that did not present barriers or facilitators perceived by users in relation to the media were excluded. Political documents, technical reports, literature reviews and studies whose full texts could not be obtained, even after attempts to contact the authors, were excluded".

About the comment "Page 8, second last para, it is suggested to describe clearly for sampling of the second search, ie. google search, that end up with 6 articles, and further adding to the main search of 15 articles"

Answer: To improve this explanation, we modified the paragraph to:

"The databases searched resulted in 1,392 studies, as shown in the flow diagram in figure 1.

Duplicate studies were removed which resulted in 1,080 articles found. All the studies found were screened by the two reviewers, and according to the eligibility criteria, a total of 134 articles were sent to the reading step of the full texts. In the selection of studies by the inclusion criteria the total number of included studies was 15 and 119 were excluded as described in the flowchart (Figure 1). In the additional strategy, which was the reading of the 300 studies found in Google Scholar in order of relevance, it was identified that 21 were duplicates that had already been found in the search of the databases. Thus, in order to have 300 studies without duplicates, 321 studies were screened. As a result, 95 studies were selected for full reading and, subsequently, 6 were included based on the same inclusion criteria applied to full texts. Therefore, the main search included 15 studies and the Google Scholar search 6 studies, this review then included a total of 21 studies."

About the comment "Limitation, ie, the limited of paper based research need to be expanded further, and how this influence the accuracy of the findings and conclusion"

Answer: This comment is very helpful and will contribute to the discussion, as it addresses a relevant point. On one hand, it demonstrates the growing use of screen-based media; however, on the other hand, it complicates the comparison between different media. Therefore, in the final paragraph of the discussion, along with another limitation, we added the following text: "Another limitation we encountered was the scarcity of studies exclusively utilising paper-based media, as none were solely dedicated for this form, and only 4 out of 21 studies found contrasted paper with screen media. Following the completed review, we identified 'printed material' as a potential descriptor that other researchers can use. While this trend might indicate the increasing use of screen media, it limits the enhancement of comparisons between the barriers and facilitators of both".

About the comment "Why accessibility of screen based has been reported as low compared to paper based need to be further expanded, with inclusion of factors such as digital illiteracy, particularly in the countries or regions where there is issues on digital accessibility etc."

Answer: We appreciate your comment, however, screen-based media was not reported as less accessible, it was discussed that, overall, it might be perceived as less accessible due to being more expensive and requiring digital literacy. In practice, it can be more accessible as it enables illiterate individuals to access information through various features, such as screen reading options or interactive videos.

Considering Reviewer 2, Dr. U Okpalauwaekwe, University of Saskatchewan

Comments to the Author:

We appreciate the following comment from the reviewer: It was a pleasure reviewing and learning about your work. The article is well-structured and provides a comprehensive review of the use of paper and screen-based media for health education among adults aged 60 and above. The authors clearly outlined the methodology, including the search strategy, data extraction, and analysis. The results are presented in a clear and concise manner, and the discussion provides a thoughtful interpretation of the findings.

Regarding the comment “The article could benefit from a more explicit statement of the research question or objective at the beginning of the paper. While it is implied that the paper is examining the use of paper and screen-based media for health education among older adults, a clear statement of the research question would help to guide the reader

Answer: To a better explanation of the objective of this review, we modified the last paragraph of the introduction to:

"Current studies, including reviews, of the use of paper or screen formats for health information are carried out separately and seldom compared with each other. Furthermore, they address a wide range of ages in the audiences involved. The aim of this review is to map the use of paper and screen based media in current health education practices targeted at older people aged 60 or over. This should allow the identification of the barriers and facilitators in the use of each media for this population, comparing the use of these two types of media for health education."

Regarding the comment “Authors have done a good job of describing their search strategy and data extraction process. However, they could provide more detail on how they assessed the quality of the included studies. For example, did they use a standardized tool for quality assessment? This would add to the rigour of the research.”

Answer: We appreciate the comment, however an assessment of the quality of the included studies was not carried out, as the objective of this scoping review was to map all studies that carried out health education for older people using screen or paper media.

Regarding this, Peters et al (2021) reports that "because scoping reviews seek to develop a comprehensive overview of the evidence rather than a quantitative or qualitative synthesis of data, it is not usually necessary to undertake methodological appraisal/risk of bias assessment of the sources included in a scoping review", in addition, the authors report that the focus of the scoping review is to describe relevant characteristics of the studies found and that this type of review can include different types of studies, such as with different methodological aspects, gray literature, among others. For the reasons mentioned above, we did not carry out an assessment of the quality of the studies included in this review.

Peters, M. D., Marnie, C., Colquhoun, H., Garritty, C. M., Hempel, S., Horsley, T., ... & Tricco, A. C. (2021). Scoping reviews: reinforcing and advancing the methodology and application. *Systematic reviews*, 10(1), 1-6.

About the comment “The discussion section could benefit from a more in-depth interpretation of the results. The authors could consider discussing the implications of their findings for practice and policy, and how their results compare to previous research in this area.”

Answer: We appreciate this comment and we added the following paragraph/sections in the discussion:

Based on the findings of our review, the discussion in the literature emphasises the significance of interface design and screen media usability. Morey et al. (2019) identified issues with an excess of icons and low contrast, highlighting participant concerns. Vanoh et al.'s (2018) study underscored that older users prefer visually appealing media featuring attractive photos and colours, while Jongstra et al. (2017) revealed their need for different functionalities. Hence, it is imperative to incorporate diverse functionalities tailored to the requirements of the older population, coupled with an engaging design characterised by colour contrast and intuitiveness to enhance usability.

The adoption of a user-centred design approach is crucial to meet the unique needs of older audiences. Co-design sessions, as suggested by Leme (2014), serve as an effective strategy for involving older individuals in the creation and design processes of new tools across various media. In this way, participatory design enables the creation of intuitive media, which, consequently, older individuals will find easy to use. Moreover, it was identified in the included studies that one of the main barriers was the difficulty of learning these screen media. Therefore, it is crucial that public policies are developed and improved for the digital inclusion of older people, such as programs that teach these individuals how to use these media, so that they can enjoy all the benefits of digital

technologies. The study by Chung et al. (2018), which aims to analyse the usability of a sleep education app for older Koreans, presents as a perceived facilitator by users the ease with which they learned how to use the application.

Additionally, functionality is identified as a barrier in the use of e-health for older adults, as noted in Wilson's (2021) scoping review. Conversely, ease of use emerged as a facilitator, as observed in both Alruwaili's (2023) and Wilson's (2021) studies, aligning with our usability review category.

Our study's results align with concerns mentioned in Olson's (2014) research, particularly regarding privacy, which is also identified as a barrier in related studies such as Alruwaili's (2023) systematic review and Wilson's (2021) study. Privacy and data security apprehensions are recognised as obstacles to digital health interventions among older adults. Therefore, it is essential to create public policies that ensure the privacy and security of users' data when using screen-based media in health education interventions. It is also important that policies regarding screen media are presented to the user at the beginning of the interaction with the media, as this will inform and ensure safety for users.

Specific Comments

Line 7-8: "The extraction table was described in the review protocol, however there was a change regarding the topic "is the health education aimed at the ... caregiver?" which was withdrawn, as the target population of the review study is adults 60+ regardless of occupation." This sentence is a bit confusing and could be clarified. It would be helpful to explain why this change was made and how it impacted the review.

Answer: The sentence was modified to:

"The extraction table was described in the protocol for this scoping review, however the topic "Is the health education aimed at the caregiver?' was withdrawn. This topic was removed from the extraction table, considering that throughout the study it was identified that the papers included were mostly focused on the participants 60+ and not caregivers".

About the comment: "Line 18-19: "In the additional strategy, which was the reading of the 300 studies found in Google Scholar in order of relevance, it was identified that 21 were duplicates that had already been found in the search of the databases, thus 321 studies were identified in order to have 300 studies without duplicates to be screened for eligibility." This sentence is quite long and could be broken up into two sentences for clarity."

Answer: This sentence was modified to:

"In the additional strategy, which was the reading of the 300 studies found in Google Scholar in order of relevance, it was identified that 21 were duplicates that had already been found in the search of the databases. Thus, in order to have 300 studies without duplicates, 321 studies were screened."

Line 34-35: "The younger generations, such as those born in 1980-2010, grew up adapting to new technologies, as they already existed when they were born." This sentence seems out of place in the paragraph and doesn't add to the discussion of the results. Consider removing or rephrasing.

Answer: Thank you for your suggestion. The sentence was removed.

Line 46-47: "Indeed, this is mentioned in the wider literature. For example Maskell, McDonald and Paudyal (2018)³⁸ studied the use of printed materials used in the waiting rooms of general clinics in England and found that they were noticed by more of three-quarters of patients, and read by a large proportion of them." This sentence could be rephrased for clarity. Consider something like: "This is supported by wider literature, such as a study by Maskell, McDonald and Paudyal (2018), which found that more than three-quarters of patients noticed printed materials in the waiting rooms of general clinics in England, and a large proportion read them"

Answer: Thank you for the suggestion. The sentence was modified to:

"This is supported by wider literature, such as a study by Maskell, McDonald and Paudyal (2018), which found that more than three-quarters of patients noticed printed materials in the waiting rooms of general clinics in England, and a large proportion read them"

Regarding the following comments: "The authors could provide more detail on how they categorised the barriers and facilitators to use. What criteria did they use to determine what constitutes a barrier or facilitator? This would add to the rigour of the research." And "Line 90-91: "Categorization of barriers and facilitators In response to the second research question about the barriers and facilitators to use, it." This sentence is incomplete and could be clarified. It would be helpful to provide more detail on how the authors categorised the barriers and facilitators to use."

Answer:

The sentence mentioned is in the text as follows: "In response to the second research question about the barriers and facilitators to use, it was identified that the vast majority were about screen-based media. Thus, they were grouped according to their similarity into 10 categories, described below". After this text, all categories and their definitions are described.

In order to explain what the barriers and facilitators were, the following phrase was added to the sentence:

"Barriers were considered to be difficulties that older people had when using screen or paper media, factors that hindered the use or the lack of features. Facilitators, on the other hand, were factors that contributed or motivated individuals to use the media."

About the comment: "The authors could provide more context when discussing the studies included in the review. For example, when discussing the study by Chung et al., 2018, it would be helpful to provide a brief summary of the study's findings and how they relate to the overall research question".

Answer: Thank you for the comment. In the studies used in the discussion, we provided a brief summary of the findings. The modified sentences were as follows:

In the twelfth paragraph of the discussion: "Regarding the use of paper media for health education, according to the findings of this review, it is clear that these media are still used, as revealed in the study by Koops et al. (2016), that used a website and a booklet to development a doctor-patient communication for older adults with limited health literacy. In this study, a booklet with photographic stories that aided in the communication between doctor and patient was considered to have good availability, as it could be given out and used anywhere".

In the sixth paragraph of the discussion: "Therefore, it is crucial that public policies are developed and improved for the digital inclusion of older people, such as programs that teach these individuals how to use these media, so that they can enjoy all the benefits of digital technologies. The study by Chung et al. (2018), which aims to analyse the usability of a sleep education app for older Koreans, presents as a perceived facilitator by users the ease with which they learned how to use the application."

In the eighth paragraph of the discussion: "In one of the studies of this review, by Pischke et al. (2022), that compared the acceptance and effectiveness of two interventions for physical activity (PA) promotion for older adults through websites, apps, brochures and diaries, they found individuals who have no affinity with screen media, preferring paper. In another study found in this review, by Peels et al. (2013), based on a perception considered an Usability barrier, that reports "The assignments integration into the web-based tailored advice were not direct and required big effort to participate", it is notable that not being intuitive leads to a greater amount of effort."

About the comment: "The authors could provide more analysis on the implications of the barriers and facilitators identified. How do these barriers and facilitators impact the use of paper and screen-based media for health education among older adults? This would add depth to the discussion section".

Answer: Thank you for the comment. The following considerations have been added:

In the eighth paragraph of the discussion: "In one of the studies of this review, by Pischke et al. (2022), that compared the acceptance and effectiveness of two interventions for physical activity (PA)

promotion for older adults through websites, apps, brochures and diaries, they found individuals who have no affinity with screen media, preferring paper. In another study found in this review, by Peels et al. (2013), based on a perception considered an Usability barrier, that reports "The assignments integration into the web-based tailored advice were not direct and required big effort to participate", it is notable that not being intuitive leads to a greater amount of effort. Thus, older people have greater difficulty with screen-based media and require more training. This training requirement is urgent at the current moment, given that the current young population, who are the future older people, are already growing up with skills and will likely have an easier time using them. (Peels et al., 2013)". In the thirteenth paragraph of the discussion: "However, for individuals who are actually illiterate, screen readers and audio content, for instance, offer a level of accessibility that paper cannot match. These functionalities are facilitators found in screen-based media seen in this review, as videos mentioned as the most useful components in the study of Olson (2014)".

About the comments: "Line 72-73: "Mobile App Use for Insomnia Self- Management in Urban Community- Dwelling Older Korean Adults: Retrospective Intervention Study 28 Chung et al., 2018 Sleep education Smartphone (app) NA." This sentence could be clarified. It would be helpful to provide a brief summary of the study's findings and how they relate to the overall research question.' Line 78-79: "The Application of Virtual Reality in Patient Education30 Pandrangi et al., 2019 Health education - health status (Website) Flyers, drawings and brochures." This sentence could be broken up for clarity. Consider something like: "Pandrangi et al., 2019 conducted a study on the application of virtual reality in patient education. The study used a website, flyers, drawings, and brochures for health education"

Line 84-85: "The Impact of 3-D Models versus Animations on Perceptions of Osteoporosis and Treatment Motivation: A Randomised Trial32 Jones et al., 2017 Information about osteoporosis Tablet (animations on Microsoft PowerPoint) NA." This sentence could be rephrased for clarity. Consider something like: "Jones et al., 2017 conducted a randomized trial studying the impact of 3-D models versus animations on perceptions of osteoporosis and treatment motivation. The study used a tablet with animations on Microsoft PowerPoint"

Answer: All these reported sentences are present in Table 2 about the studies included in the review. This table presented the following information: the title of the study, the authors and the year, the central theme of the health education and the screen and paper-based media format was used (the screen media used and the paper media was not used so it was written NA (explained in the table caption as not applicable).

Therefore, the table only presents this information. More details of the included studies are described after the table.

We would like to clarify that the data included in the tables, as well as in the overall results and discussion, are those that address the research questions. Table 2 contains the studies and their respective media types and themes. Meanwhile, in the remaining results and discussion sections, we outline the identified barriers and facilitators, as well as the comparison between the usage of the two media types when such data was available in the study.

VERSION 2 – REVIEW

REVIEWER	Okpalauwaekwe, U University of Saskatchewan, Medicine
REVIEW RETURNED	12-Feb-2024
GENERAL COMMENTS	Thank you for the opportunity to review your work. The study entitled "Design considerations for the migration from paper to screen media in current health education for older adults: A scoping review" intends to map the use of paper-based and

	screen-based media for health education targeted at older adults. The study is structured around a scoping review using the PRISMA-ScR guidelines, and a search strategy across multiple databases and languages, supplemented by grey literature search through Google Scholar. General Comments: The topic is relevant and addresses an important aspect of health education delivery in an aging population. The manuscript appears to be well-structured, following a systematic approach to literature review and data analysis. The diversity in data sources and the multilingual approach enhance the robustness of the findings, and the inclusion of grey literature is commendable as it offers a broader perspective beyond peer-reviewed articles. Specific Comments: Introduction:  1. Provides a clear rationale for the research, highlighting the demographic shift and its implications for health education. 2. Good use of statistics to emphasize the importance of the study subject. Well done, as it sets a solid background for the study objectives. Methods:  1. Detailed and thorough, following recognized guidelines and frameworks. 2. The search strategy and selection process are well-articulated, ensuring reproducibility. 3. The inclusion and exclusion criteria are clear, aiding in understanding the scope of the review. Results:  1. Results are presented in a structured manner with the use of tables for clarity. 2. The flow of information from the search results to the final study selection is well-documented. 3. The categorization of barriers and facilitators provides a detailed analysis of the findings. Discussion: The discussion offers insights into the implications of the findings, linking back to the objectives and the introduction. It reflects on the transition from paper to screen media and the design considerations that could influence this migration. The section could benefit from a deeper exploration of how these findings relate to existing literature. Conclusions: The conclusions drawn are appropriate and align with the study's findings. Recommendations for future media design and the suggestion for combined use of media types are practical and forward-thinking. Strengths and Limitations:
--	--

	The manuscript acknowledges its own limitations, which is essential for a balanced view. It would be beneficial to explore potential biases introduced by the chosen methodology or the studies included. All the best!
--	--

VERSION 2 – AUTHOR RESPONSE

Considering Reviewer 2: Dr. U Okpalauwaekwe, University of Saskatchewan

We appreciate all the comments about the information that the review presents in all its sections. About the comment: “It would be beneficial to explore potential biases introduced by the chosen methodology or the studies included.” We included the following text in the limitations: “In addition, a methodological bias was the inclusion criteria that participants should have reported barriers or facilitators, which resulted in a reduction in the number of articles included. Furthermore, as it is a scoping review, its objective is to map all studies on the topic, therefore, an assessment of the quality of the included studies was not carried out.”